# Successful delivery of large-size CRISPR/Cas9 vectors in hard-to-transfect human cells using small plasmids

Jonas Nørskov Søndergaard[1], Keyi Geng[1,3], Christian Sommerauer[1,3], Ionut Atanasoai[1], Xiushan Yin[1,2] & Claudia Kutter[1✉]

With the rise of new powerful genome engineering technologies, such as CRISPR/Cas9, cell models can be engineered effectively to accelerate basic and disease research. The most critical step in this procedure is the efficient delivery of foreign nucleic acids into cells by cellular transfection. Since the vectors encoding the components necessary for CRISPR/Cas genome engineering are always large (9–19 kb), they result in low transfection efficiency and cell viability, and thus subsequent selection or purification of positive cells is required. To overcome those obstacles, we here show a non-toxic and non-viral delivery method that increases transfection efficiency (up to 40-fold) and cell viability (up to 6-fold) in a number of hard-to-transfect human cancer cell lines and primary blood cells. At its core, the technique is based on adding exogenous small plasmids of a defined size to the transfection mixture.

[1] Department of Microbiology, Tumor, and Cell Biology, Karolinska Institute, Science for Life Laboratory, Solna, Sweden. [2] Applied Biology Laboratory, Shenyang University of Chemical Technology, Shenyang, China. [3]These authors contributed equally: Keyi Geng, Christian Sommerauer. ✉email: claudia.kutter@ki.se

CRISPR/Cas has revolutionized genome engineering of biological systems due to its easy design, target site specificity, and scalability for high-throughput applications. It allows gene deletions, enhancing or inhibiting gene expression in vitro and in vivo. The components of the CRISPR/Cas system (including guide RNAs) are often encoded on large extrachromosomal expression vectors (9–19 kb) that are delivered into cells via transfection. Due to the large size, these vectors are notoriously difficult to transfect and cause high cell death, which prohibit downstream analyses[1,2].

Cell transfection method development has resulted in safer viral vectors (biological)[3], new polymers and lipids (chemical)[4], and particle delivery devices (physical)[4]. Viral-mediated delivery (transduction) leads to the highest efficiencies but requires higher biosafety level laboratory settings and ethical approval when used in research or in the clinic[5]. We overcame limitations of current electroporation-based transfections by adding appropriate amounts of small (~3 kb) to large (9–15 kb) vectors, which increased transfection efficiency and cell viability. Due to its easy implementation in current transfection protocols, this strategy may be broadly applicable in basic and applied research.

## Results

**Small vectors improve transfection efficiencies.** Standard transfection via electroporation (Fig. 1a, Table 1) of a 15 kb CRISPR-GFP vector into hard-to-transfect human lung cancer cells (A549) showed extremely low transfection efficiency (4.2%) and high cell death (91%) (Fig. 1b, c). In contrast, co-transfection of equal mass of a small empty vector (3 kb) together with the large CRISPR-GFP vector (15 kb) drastically increased transfection efficiency (40%) and reduced cell death (45%) (Fig. 1b, c).

We next tested if the size of the small vector influences transfection efficiencies by using a range of small vectors (1.8–6.5 kb) (Fig. 1d, Supplementary Fig. 1, Table 2). On average, co-transfection of the large CRISPR-GFP vector with small vectors increased transfection efficiency by 12.2% (4.9-fold change) and cell viability by 16.6% (1.9-fold change). Of all vectors tested, the small vector of 3 kb showed the highest increase in transfection efficiency (average of 21.4% or 6.8-fold increase) (Fig. 1d, Supplementary Fig. 1f–h). Furthermore, transfection of solely small vectors did not significantly alter cell viability when compared to mock transfection (paired two-tailed t-test, p-value > 0.05) (Supplementary Fig. 2). Since all small vectors improved transfection efficiencies, we speculate that our approach has been

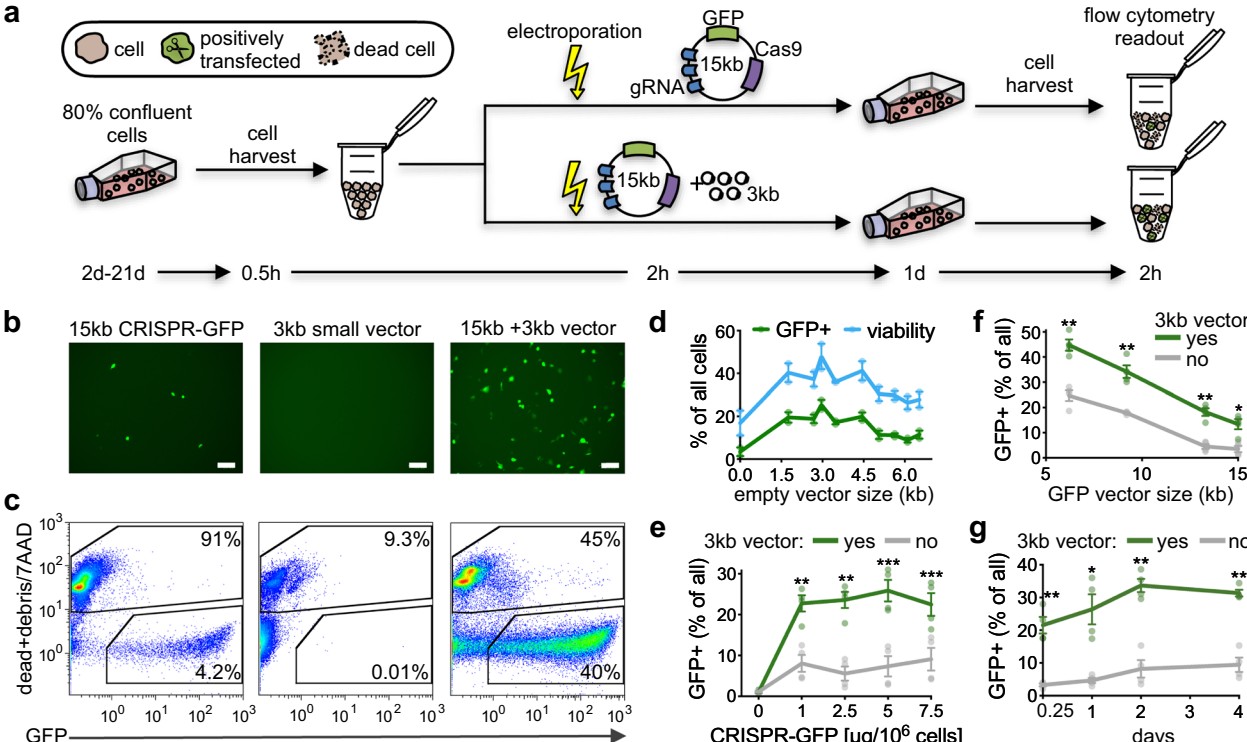

**Fig. 1 Transfection efficiency can be improved by co-transfecting large CRISPR vectors with small vectors. a** Schematic overview of the cell transfection setting. Electroporation-mediated transfection (lightning bolt) of a large CRISPR-GFP vector (15 kb) without (above) and with (below) a small vector (3 kb). Duration in days (d) and hours (h) for each experimental procedure is indicated. **b, c** Microscopy images and flow cytometry plots (gating of GFP+ and 7AAD dead-cell marker) of hard-to-transfect A549 cells 24 h after electroporation (left: 15 kb CRISPR-GFP vector alone, middle: 3 kb small vector alone, right: co-transfection of 15 kb CRISPR-GFP and 3 kb small vector). Scale bar: 100 μm. Amounts of vector and electroporation conditions can be found in Table 1. **d** Line graph illustrates percent transfection efficiency (green) and cell viability (blue) upon co-transfection of a large 15 kb vector with small vectors of varying sizes (1.8–6.5 kb) in A549 and MCF7 cells (n = 4, mean ± SEM). **e** Line graph demonstrate the percent transfection efficiency after co-transfection of a large CRISPR-GFP vector (15 kb) of varying concentrations without (gray) and with (green) a small vector (3 kb) in A549 and MCF7 cells (n = 4, mean ± SEM). **f** Line graph demonstrate the percentage of viable GFP + cells after co-transfection of large GFP vectors (6.5–15 kb) without (gray) and with (green) a small vector (3 kb) in A549 and MCF7 cells (n = 4, mean ± SEM). **g** Line graph demonstrate the percent transfection efficiency after co-transfection of a large CRISPR-GFP vector (15 kb) without (gray) and with (green) a small vector (3 kb) in A549 and MCF7 cells from 6 h (0.25d) to 4d after transfection (n = 4, mean ± SEM). Statistics: paired two-tailed t-test, *p < 0.05, **p < 0.01, ***p < 0.001.

**Table 1 Electroporation conditions used in each experiment (unless otherwise stated).**

| Cells | # of cells | µg GFP vector | µg small vector | Voltage | ms | pulses | Further optimized from manufacturer's settings |
|---|---|---|---|---|---|---|---|
| Huh7 | $10^6$ | 7.5 | 7.5 | 1000 | 40 | 2 | Yes |
| HepG2 | $10^6$ | 2.5 | 7.5 | 1200 | 30 | 2 | Yes |
| A549 | $10^6$ | 5 | 5 | 1230 | 30 | 2 | No |
| HEK293 | $10^6$ | 5 | 5 | 1100 | 20 | 2 | No |
| MCF7 | $10^6$ | 5 | 5 | 1100 | 30 | 2 | No |
| HL60 | $10^6$ | 5 | 5 | 1350 | 35 | 1 | No |
| PC3 | $10^6$ | 5 | 5 | 1450 | 10 | 3 | No |
| SH-SY5Y | $10^6$ | 5 | 5 | 1200 | 20 | 3 | No |
| PBMCs | $10^6$ | 5 | 5 | 2150 | 20 | 1 | No |
| CD8 T cells | $10^6$ | 5 | 5 | 2100 | 20 | 1 | No |

**Table 2 Number of molecules of the different small vectors used in Fig. 1d.**

| Small vector | size [kb] | mass [µg] | molarity [pmol] |
|---|---|---|---|
| pUC19-no-LacZ | 1.757 | 5 | 4.605 |
| pUC19 | 2.686 | 5 | 3.012 |
| pBlueScript | 2.961 | 5 | 2.733 |
| pH6HTC | 3.473 | 5 | 2.330 |
| pH6HTC-STMN | 4.448 | 5 | 1.819 |
| pH6HTC-PKM | 5.057 | 5 | 1.600 |
| pH6HTC-CCT | 5.633 | 5 | 1.436 |
| pH6HTC-CTCF | 6.209 | 5 | 1.303 |
| pH6HTC-D9 | 6.531 | 5 | 1.239 |

employed unknowingly when co-transfecting gRNAs and CRISPR/Cas components on separate vectors[6]. Transfection of cells with increasing amounts of CRISPR-GFP vector (15 kb, 0–7.5 µg) did not result in an increase of GFP+ cells but rather lead to a higher number of dead cells (Fig. 1e, Supplementary Fig. 3). Co-transfection of a fixed amount of the small vector (3 kb, 5 µg) increased the number of GFP+ cells consistently (4.3-fold change on average) and increased the number of viable cells (1.9-fold change on average) (Fig. 1e). This suggests that the size but not the amount of the large vector affect transfection efficiencies.

To validate whether the increase in transfection efficiency and cell viability was dependent on the size of large CRISPR vectors, we electroporated cells with a range of different GFP vectors (6.5–15 kb). We found a gradual decrease in transfection efficiency (from ca. 25 to 4%) and cell viability (from ca. 36 to 15%) with increasing GFP vector size (Fig. 1f, Supplementary Fig. 4). This vector size-dependency on transfection efficiency can be considered when designing future CRISPR vectors. Remarkably, transfection efficiency and cell viability improved considerably upon co-transfection of the small vector (3 kb) with all tested GFP vectors. Overall, we found an average increase in transfection efficiency of 15% (range: 6–25%) irrespective of the size of the GFP vector. Additionally, GFP expression in cells that were transfected with a large CRISPR-GFP vector (15 kb) in the presence of small (3 kb) plasmids showed a stable enhancement in transfection efficiencies lasting for several days (6 h to 4d after transfection) (Fig. 1g, Supplementary Fig. 5).

**Small vectors increase transfection efficiency in numerous cell types.** Patient-derived human cell lines have been intensively used in research to investigate molecular mechanisms explaining diseases as well as to identify and test pharmaceutical compounds for therapeutic purposes under well-defined and reproducible conditions. Furthermore, primary cells from the peripheral blood

isolated from patients can be genetically modified and used as immunotherapy in clinical applications. In order to determine whether our approach can be employed in other intensively studied, hard-to-transfect or primary cell types, we co-transfected small (3 kb) and large (15 kb) vectors to measure transfection efficiencies and cell survival in diverse adherent and non-adherent cancer cell lines (Huh7 and HepG2 (liver), PC3 (prostate), MCF7 (breast), HEK293 (kidney), A549 (lung), SH-SY5Y (neuronal), HL-60 (leukaemia)) as well as peripheral blood mononuclear cells (PBMCs) and purified CD8 T cells (Fig. 2a). In line with our previous observations, we confirmed a consistent improvement of transfection efficiency (up to 36%) and cell viability (up to 46%) in all tested cell lines (Fig. 2b–e, Supplementary Fig. 6). In contrast to adherent cells, cells in suspension had a very low transfection efficiency, which could probably be improved upon further optimization of the electroporation settings. However, co-transfection with the small vector did nevertheless improve the number of GFP+ cells (Fig. 2b, d, Supplementary Fig. 6) but did not increase cell viability (Fig. 2c, e). Overall, we noticed that the increased percentage of positively transfected cells correlated highly with the percentage of viable cells (Fig. 2f), suggesting that the mode of action of the small vector is to improve viability of positively transfected cells. To better understand the underlying mechanism, we examined dependencies of transfection efficiencies on the small vector conformation (Supplementary Fig. 7a–e). Linearizing the small vector did not lead to any difference in transfection efficiency when compared to its circular version, suggesting that circular and linearized plasmids are equally capable of enhancing the transfection efficiency. We also considered vector sequence content and features and found that neither GC content nor specific motifs in the encoded DNA of the small vector improved transfection efficiencies (Supplementary Fig. 7f).

Finally, we tested our approach in chemical transfection methods. We employed the commonly used Lipofectamine 3000 and generated liposomes containing the large CRISPR-GFP vector (15 kb) in the presence and absence of a small vector (3 kb). Similar to the results obtained after electroporation, the addition of the small vector enhanced transfection efficiencies in all of the tested cell types (Supplementary Fig. 8). However, in contrast to electroporation-based transfection, inclusion of the small vector in the liposomes slightly decreased the viability (Supplementary Fig. 8).

## Discussion
We postulate that small vectors can rapidly pass cell and nuclear membranes and large vectors can move on this flow of small vectors into the cell. By co-travelling with small vectors, larger vectors may enter the cell without getting entangled in two or more open membrane pores, which ensures proper plasmid uptake and

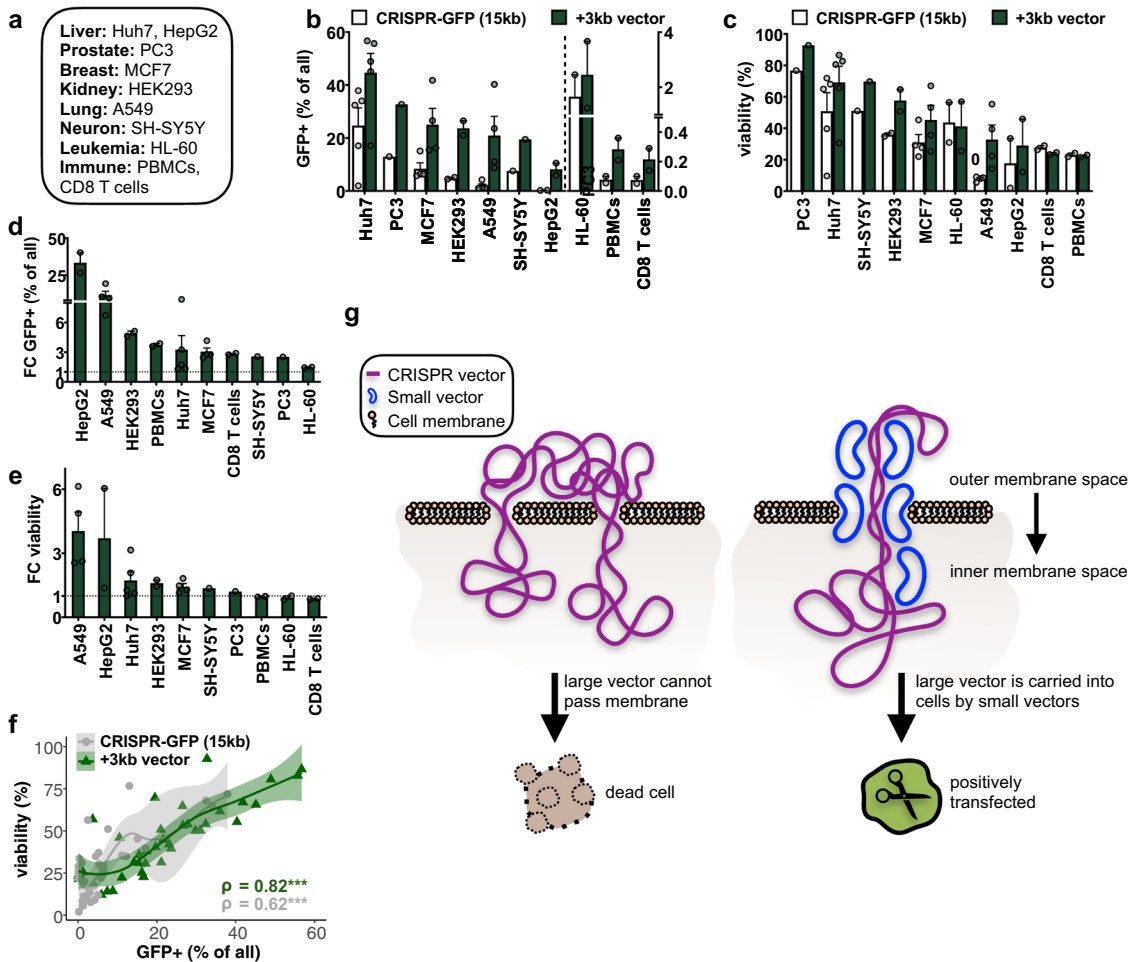

**Fig. 2 Co-transfection of small vectors increases transfection efficiency in numerous cell types. a** Various human cancer cell lines and freshly isolated primary immune cells were electroporated. **b, c** Bar graphs depicting the percent increase of **b** transfection efficiency and **c** cell viability upon co-transfection of the CRISPR-GFP (15 kb) vector without (white) or with (dark green) a small vector (3 kb) in the tested adherent (left of the dotted line) and non-adherent (right of the dotted line) cells ($n = 1$–6, mean + SEM). Cell types are ranked by decreasing transfection efficiency or cell viability after adding the small vector. Amounts of vector and electroporation conditions can be found in Table 1. **d, e** Bar graphs illustrate the fold change (FC) in the number of **d** GFP+ and **e** viable cells after adding a small 3 kb vector ranked by FC-enrichment in the tested cells. **f** Plot shows Spearman's rank correlation coefficients ($\rho$) and $p$-values (***$p < 0.001$) of transfection efficiencies ($x$-axis) and cell viability ($y$-axis) (in percent) without (gray) or with (green) co-transfection of a small vector (3 kb) ($n = 26$, 95% confidence interval). **g** Model explaining increased transfection efficiencies of large CRISPR vectors (purple) upon addition of small vectors (blue). Membranes and nuclear pores are coated by small vectors and thereby facilitates efficient delivery into the nucleus and subsequent molecular activity of the large vector.

membrane reclosure thereby preventing cell death[7] (Fig. 2g). This model would explain the results we have obtained using electroporation of adherent cells, but it explains neither the poor viability of suspension cells nor why the small vector improves liposome transfection efficiency. Compared to the adherent cells, cells in suspension are generally smaller and have not undergone trypsinization. Trypsinization cleaves the bonds that stretch the membrane of the adherent cells and may thus result in a less compact and a more relaxed membrane structure. This may facilitate the formation of larger membrane pores after electroporation, allowing better accessibility of small plasmids and thereby smoother membrane passage of larger vectors (Fig. 2g). The difference in cell death after co-transfection with small vectors may be further explained by different responses of DNA sensors triggering programmed cell death[8]. DNA sensors might be inert in response to small plasmids but highly active when sensing large plasmids. However, this phenomenon is speculative, and more biophysical work is required to resolve the exact underlying mechanism.

An alternative approach to our strategy has previously been published[2,9]. Unlike transcribing the CRISPR/Cas9 system inside the host cell as we propose, a recombinant CRISPR/Cas9 ribonucleoprotein can be formed in vitro prior to electroporation. This approach yielded higher transfection efficiencies in primary blood cells than we report. However, a recombinant protein has to be produced and the ribonucleoprotein complexes must be electroporated immediately after formation. Compared to this strategy, our approach of adding a small vector to the transfection mixture is simpler, cheaper and less time-consuming.

In summary, we discovered that electroporation and lipofectamine-based cell transfection of cancer cell lines and primary cell types can be improved by adding a small vector to the transfection mixture. As CRISPR technology is universally applicable and will continue to develop further, our optimized, simple and non-hazardous transfection approach will have numerous applications in clinical biomedicine and industrial biotechnology.

## Methods
**Cell culture and cell lines.** HepG2, Huh7, PC3, SH-SY5Y, HEK293, MCF7, HL-60, and A549 cancer cell lines were obtained from American Type Culture

Collection (ATCC). All cell lines were mycoplasma-free when periodically tested with Mycoplasmacheck (Eurofins Genomics) or MycoProbe (R&D). Cells were cultured in T-75 flasks at 37 °C and 5% $CO_2$ atmosphere using medium supplemented with 1/100 Penicillin/Streptomycin (Sigma) and 10% fetal bovine serum (Hyclone). Huh7, HepG2, A549, HEK293, and MCF7 were cultured in Dulbecco's Modified Eagle Medium (DMEM, Sigma), HL-60 was cultured in Roswell Park Memorial Institute (RPMI) 1640 (Sigma), and PC3 and SH-SY-5Y were cultured in DMEM:F-12 (1:1) medium (Gibco). To ensure authenticity, cell lines were initially genotyped by short-tandem repeat genetic profiling (STR) using the Power-Plex_16HS_Cell Line panel and analyzed using Applied Biosystems Gene Mapper ID v3.2.1 software by the external provider Genetica DNA Laboratories (LabCorp Specialty Testing Group) and continuously assessed phenotypically. Cells were split at ~70–90% confluency by aspirating the medium, gently washing with phosphate buffered saline (PBS, Sigma) and detaching them with 3 mL of a trypsin-EDTA solution (Sigma) for 3–5 min. Trypsin was inactivated with a minimum of 10-fold surplus of culture medium before a cell fraction was passaged.

**Vectors**. pLV hU6-sgRNA hUbC-dCas9-KRAB-T2a-GFP[10] (15.0 kb), FC3-GFP (6.2 kb), SpCas9(BB)-2A-GFP-prickle (9.2 kb), pCAGGs-jmj1dc-IRES-GFP (13.3 kb), pUC19-no-LacZ (1.8 bp), pUC19 (2.7 kb), pBlueScript (3.0 kb), pH6HTC (3.5 kb), pH6HTC-STMN (4.4 kb), pH6HTC-PKM (5.1 kb), pH6HTC-CCT (5.6 kb), pH6HTC-CTCF (6.2 kb), pH6HTC-D9 (6.5 kb).

**Electroporation**. Cells were electroporated using the NEON electroporation system (Invitrogen). Briefly, cells grown to 70–90% confluency were harvested and pelleted at 500 g for 5 min at room temperature. Cells were resuspended in PBS, counted, and spun down at 500 g for 5 min at room temperature. The appropriate amount of plasmid DNA (see Table 1) were transferred into a sterile 1.5 mL microcentrifuge tube. After aspirating PBS from the cell pellet, the cells were resuspended in Resuspension Buffer R to $1.0 \times 10^7$ cells/mL. Cells were gently mixed to obtain a single cell suspension and added to the tube containing plasmid DNA. The cells were mixed gently with the plasmids without creating any air bubbles. To avoid unnecessary cell death, the electroporated cells were directly plated into a pre-heated phenol red-free medium without any antibiotics. Electroporation settings used are found in Table 2.

**Liposomal transfection**. Liposomal transfection was done with lipofectamine 3000 according to the manufacturer's instructions. All cells were plated in 24-well plates (110,000/well HepG2 and MCF7, 70,000/well Huh7, and 50,000/well A549). Mix 1: 25 μL Opti-MEM (Gibco) + 1.5 μL Lipofectamine 3000. Mix 2: 25 μL Opti-MEM + 250 ng pBluescript + 250 ng GFP-vector + 1 μL P3000 reagent.

**Transfection efficiency and cell viability measures**. Transfection efficiency was measured after 24 h (unless stated differently) by flow cytometry (FACSNavios, Beckman Coulter, Navios Cytometry List Mode Data Acquisition and Analysis Software version 1.3) by gating cells for GFP and 7AAD (FlowJo version 8.2, Supplementary Fig. 1). In order to collect all potentially dead cells, both the supernatants and the adherent cells (harvested by trypsinization) were collected. Cells were washed in 1xPBS with 1% BSA (Sigma), followed by staining in 100 μL buffer with 5 μL 7AAD viability staining solution (eBioscience) for 15 min on ice in the dark. Cells were acquired directly without washing away the staining buffer by flow cytometry. In some experiments, microscopy images were taken using a Zoe Fluorescent Cell Imager (Bio-Rad, software version 002.257.011215).

**Primary immune cell isolation**. Buffy coats were obtained from anonymous healthy volunteers after informed consent and according to institutional guidelines (Karolinska University Hospital, Stockholm, Sweden). PBMCs were isolated using Ficoll-Paque (GE Healthcare) as previously described[11]. CD8 T cells were isolated using magnetic associated cell sorting (Miltenyi) according to the manufacturer's instructions. PBMCs and CD8 T cells were rested overnight at 37 °C and 5% $CO_2$ atmosphere in RPMI supplemented with 1/100 Penicillin/Streptomycin and 10% fetal bovine serum prior to electroporation.

**Statistics and reproducibility**. The data were plotted and analysed with GraphPad Prism (version 8.4.2), Microsoft Excel (version 16.37), R (version 3.6.1), and ggplot2 (version 3.3.0). Replicates are defined as individual passages of cancer cell lines or as individual donors for primary immune cells. Data are represented as mean + or ± SEM and sample sizes are presented in each figure legend. Statistical significance of differences among groups was determined by paired two-tailed student's t-test. A p-value smaller than 0.05 ($p < 0.05$) was considered to be statistically significant.

**Reporting summary**. Further information on research design is available in the Nature Research Reporting Summary linked to this article.

## Data availability

All data generated or analyzed during this study are included in the article. Raw data for graphs can be found in Supplementary Data 1. Additionally, all relevant data are available from the authors upon request.

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

## Acknowledgements

We would like to thank the KI Innovation office, specifically Åsa Kallas and Mark Didmon for helpful discussions. This work was supported by the Knut & Alice Wallenberg foundation (KAW 2016.0174, C.K.), Ruth & Richard Julin foundation (2017–00358 and 2018–00328, CK); SFO-SciLifeLab fellowship (SFO_004, C.K.), Swedish Research Council (2019–05165), Chinese Scholarship Council (KG, C.K.), KI-KID funding (2016–00189 and 2018–00904, C.K.), and the Nilsson-Ehle Endowments (J.N.S.). Open access funding provided by Karolinska Institute.

## Author contributions

J.N.S., X.Y., and C.K. conceptualized the project. J.N.S., K.G., C.S., and I.A. performed the laboratory experiments. J.N.S. did the analysis and visualized the data. J.N.S. and C.K. acquired the funding. J.N.S. and C.K. wrote the original draft. All authors contributed to the review and editing process.

## Competing interests

The authors declare no competing non-financial interests but the following potential competing financial interest: a patent has been filed for the use of small vectors to increase cellular uptake of large vectors upon transfection. Applicant: Biotech & Biomedicine (Shenyang) Group Ltd. Authors: Claudia Kutter, Jonas Nørskov Søndergaard, and Xiushan Yin. Application number: 202010315646.4. Status of application: submitted, first instance. Submitted to: National Intellectual Property Administration, People's Republic of China. Besides from this patent application, the authors declare no additional competing financial interests.
