## [Peer Review File · Communications Biology]

Reviewers' comments:

Reviewer #1 (Remarks to the Author):

This manuscript highlights the difficulties associated with delivering large DNA constructs by electroporation or chemical-based transfection. The authors demonstrate increased toxicity and reduced transfection efficiency as DNA length increases, a well-described phenomenon that poses significant challenges for several genome editing applications including the introduction of large multicistronic plasmids, as in the current study, but also with relevance for other plasmids of interest [1, 2]. The current manuscript suggests this effect may be partially overcome by co-transfection with plasmids of smaller size, with optimal benefit seen at a size of ~3 kb. While this would be an interesting and potentially useful finding, there are several technical limitations to the flow cytometric analysis and points of confusion with the described data that limit interpretation. A re-evaluation of the data to address these issues along with supplemental studies to identify the optimal DNA concentration used for transfection would help support the author's conclusions. In addition, the significance of the findings would be strengthened by demonstration of genome editing outcomes, especially given the author's focus on CRISPR-based genome editing. Finally, a discussion of alternative delivery methods such as RNP electroporation combined with a more complete discussion of the proposed mechanism would help frame this work.

Major points:

1. There are several issues with the current flow cytometric analysis. First, the bulk of the provided data is generated by gating the percentage of 7AAD+ events ("dead cells") versus the percentage of GFP+7AAD- events ("GFP+ cells") on a single plot, as exemplified in figure S1c. This strategy links these two populations so that when one goes up, the other must go down. In conditions with low viability, as in the left-most panel of figure 1d, the transfection efficiency will therefore appear artificially low because almost all events fall in the dead cell gate. This linkage may at least partially explain the strong positive correlation shown in figure 2f. This issue can be overcome by measuring transfection efficiency as a percentage of live cells rather than total events. Second, the gating strategy includes a large population of cellular debris. This is visible as a second population with low forward-scatter in the example gating shown in figure S1a-b, and as multiple populations visible in the 7AAD+ gate of all example plots (e.g. figure 1d, S1c). Inclusion of this debris further confounds the current analysis, as each of these events contributes to the percentage of "dead cells" and, conversely, takes away from the percentage of "GFP+ cells". This would be expected to overestimate the percentage of dead cells and further underestimate the transfection efficiency. Finally, the question of cellular viability is not adequately addressed by quantifying the relative percentage of live/dead cells at a single time point - this number will vary with mechanism of toxicity, time-course, proliferation, media change, and other factors. A more appropriate measurement is the number of live cells in each condition versus the number of live cells in an untreated control (which can be expressed as a percentage), or in the absence of an untreated control, directly comparing the absolute number of live cells between conditions. The 1st reference in this paper provides several excellent examples for addressing all of these concerns[1]. Specifically, please consider the following changes:
 - a. Remove the population of low forward-scatter debris from all subsequent analyses. (Left-most population in figure S1a). Although debris could be related to toxicity, including these points confuses measurements of efficiency.
 - b. Evaluate transfection efficiency as percentage of GFP+ live cells (see Ref [1]- figure 1b and section "Flow cytometric analysis an measurement of viability"). An absolute count of GFP+ live cells should also be included to evaluate the absolute number of transfected cells.
 - c. Evaluate viability as the percentage of live cells versus an untreated control, or in the absence of an untreated control, directly comparing the absolute number of live cells between conditions (see Ref [1]- figure 1a and section "Flow cytometric analysis an measurement of viability").

2. Some of the labeling and experimental details are unclear.

a. For figures 1e and S2f, show data as absolute % with and without plasmid, rather than absolute % increase, which is confusing to interpret as written. Please also incorporate changes described in point 1.

b. Please indicate the amount of each plasmid being added for each experiment (mass or moles). These details are essential for interpreting the data.

c. There are several points of confusion with the liposomal transfection experiments. The main text suggests similar benefits were seen with electroporation and liposomal-based transfection, supported by figure S3. As labeled, this figure appears to show that inclusion of the 3kb plasmid leads to increased transfection efficiency (figure S3c) but decreased viability (figure S3d). This contrasts with the increased viability seen with electroporation-based experiments. In addition, the explanatory text for figure S3 indicates that the 15kb "with" 3kb plasmid data should be on the left rather than right for figure S3c-d. Finally, it is unclear which of the comparisons in figure S3c-d the statistical significance is meant to apply to. Please consider the following changes:

i. Correct the labeling and description for figure S3 to harmonize and accurately reflect data.

ii. If the current labeling of figure S3c-d is accurate, highlight the different results seen with electroporation and lipofection in the main text and provide some interpretation of these differences in your discussion.

iii. Indicate which comparisons the statistical significance applies to in figure S1c-d.

3. The toxicity and transfection efficiency of a given DNA construct are highly sensitive to variations in DNA concentration [1, 2]. An alternative explanation for the effects seen in this study is that the molar excess of small plasmid outcompetes the larger plasmid, lowering the effective concentration and subsequent toxicity. If this example were true, one could achieve a similar effect by simply lowering the concentration of the large DNA construct. To address this concern, please first perform a titration to determine the optimal DNA concentration for viability and transfection efficiency in each cell line. An example of such an analysis is shown in figure 3 of Ref [1]. Next determine whether inclusion of the small plasmid at this optimal concentration shows the same benefit.

4. The claims regarding consistent improvement in primary cells lines are not adequately demonstrated. From figure 2b, the maximum transfection efficiency appears to be ~2.5% for PBMCs and ~1% for CD8+ T cells with no difference in viability. While transfection efficiency is not directly comparable to genome editing outcomes (which were not examined, see point #7), knockout rates of >90% are routinely achieved with electroporation of pre-assembled Cas9:gRNA ribonucleoproteins (RNP) in these cell types. Within this context, the numbers shown here would not be considered significant. Please rephrase claims regarding primary cells and provide discussion of alternatives such as RNP electroporation [2].

Minor points:

5. The discussion of potential mechanisms is interesting but incomplete. What are the primary causes of DNA-mediated cellular toxicity and how would these be affected by DNA length? Is there evidence that entanglement in multiple pores is a major factor reducing transfection efficiency or viability? How would a 3kb plasmid reduce multi-pore entanglement of a 15kb plasmid? Could this mechanism also explain results seen with lipofection-based approaches? Please consider a more comprehensive discussion to frame these results and suggest future studies.

6. Most mammalian expression plasmids that incorporate Cas9, a selectable marker, and gRNA are in the 9kb range (<https://www.addgene.org/crispr/mammalian/>). The larger plasmids used here contain additional elements for viral packaging and integration that would only be relevant for use in packaging cell lines. While this difference may not change the results, it could be confusing or misleading for readers. Please adjust text to accurately reflect the available choices for plasmid-based

delivery. For example, instead of 12-19kb, you could say 9-19kb or up to 19kb.

7. There is an emphasis on CRISPR/Cas genome engineering throughout the paper, and yet there is no analysis of genome editing outcomes. If CRISPR/Cas9 is the example use-case, it would be beneficial to show how this method affects some readout of Cas9-mediated editing (e.g. indel formation, knockout efficiency, homology-directed-repair, etc.). In addition, there is no discussion of alternative methods, such as RNP electroporation, that are widely used and provide substantial increases in editing efficiency for many of the cell types included in this study [2].

References:

1. Lesueur, L.L., L.M. Mir, and F.M. Andre, Overcoming the Specific Toxicity of Large Plasmids Electrotransfer in Primary Cells In Vitro. *Mol Ther Nucleic Acids*, 2016. 5: p. e291.
2. Roth, T.L., et al., Reprogramming human T cell function and specificity with non-viral genome targeting. *Nature*, 2018. 559(7714): p. 405-409.

Reviewer #2 (Remarks to the Author):

Communications Biology Review
12/30/19

This manuscript contains a very interesting observation that the addition of “small” plasmids seems to increase transfection (done by any of several tested methods, although the significance for methods other than electroporation are unclear) of the very large DNA vectors needed to deliver CRISPR/Cas-encoding plasmids into several different experimentally relevant cell types. This very short and somewhat superficial manuscript is written well, and the figures are mostly clear.

The data shown appear to be done fairly well. The range of plasmid lengths is fairly extensive. The different cell types tested are also extensive. The observation is important and useful to the field and beyond. The phenomenon shines a light on just how little we understand the process of transfection. It also highlights that transfection has been optimized for in vitro work over a short period of time (here 24 hours). How cell death changes with time may be important but is ignored.

There are several issues related to quantitative evaluation that are lacking in the current manuscript:

1. Time following electroporation/transfection strongly affects cell death and maybe also transfection efficiency and varies per cell type. Both GFP fluorescence and cell viability were analyzed at 24 hours post transfection, but this one time point may be too short (or too long) to capture cell death for some cell types.
2. What is the half-life of the GFP (which GFP variant is this?) encoded by the “15kb CRISPR-GFP vector?”
3. It seems that an important control experiment is missing of adding the equivalent increase of DNA mass (regardless of length) of just the 15kb plasmid to see whether increased DNA increases transfection and decreases cell death or whether the increased mass must only come from smaller plasmids. One would think that the effect of DNA length test done (shown in Fig. 1F) would test this possibility, but the same mass of 15 kb vector was added over each experiment and the same mass of “smaller” plasmid.
4. Would NUMBER (rather than mass) of plasmids give the same result? I would like to see a column added to the table shown in the methods where moles of each of the “smaller” plasmids are also

shown. The authors do not consider this possibility.

5. In Fig. 1a, the schematic showing transfection in tubes is overly simple. A schematic showing transfection, incubation, trypsinization, and analysis and including the temporal aspect of each of these steps is more useful than an overly detailed first step only.

6. In Fig. 1c and S1h, please add size bars and define them in the legend.

7. In Fig. 1e, the lines drawn are clearly not "best fit." How were these data fit? If drawn merely as a visual guide, please state so. And do not fill in between "0" and the shortest length plasmid tested. One should never infer data not done.

8. In Fig. 2, there are several issues: (a) the "legend" for g that appears above the drawing would be improved if rather than a vector in purple the exact same length as the "small plasmid" being depicted for "CRISPR vector," instead just put a purple line. These two circles are not the same size, as indicated wrongly in the figure. (b) the sideways lines on the right side are not defined (and clutter the image). At first, I thought perhaps they were meant to convey increased negative charge (-) but that they had all turned sideways erroneously, but that is not a model mentioned, so it must mean something else. (c) the "cell membrane" shown in G is confusing. In the text, the authors mention a model of the DNA entering nuclear pores. Is this actually the nuclear envelope instead of "cell membrane" as labeled? Either way, the model is not well thought through. If it is a cell membrane and what is shown is a small part of a larger cell, I do not understand why the membrane "ends" are rounded where no DNA is shown entering the cell. Is what is shown the electroporation case and there are holes (with no DNA)? Are we to assume that the bilayer membrane is depicted as the gray line where it is attached and therefore what is depicted is the whole cell or the whole nucleus? If it is supposed to be the nucleus, then the actual nuclear pores, which are not just discontinuations of nuclear envelope, should be shown but then "cell membrane cannot reseal" as a mechanism for cell death doesn't make sense.

9. The net difference in length between the large 15kb and the small 3 kb plasmids is only 5-fold (15000 compared to 3000), but in Fig. 1B and Fig. 2, the small circle is drawn ~300-fold smaller. Furthermore, the number of small circles shown is at ~20:1 ratio to the large plasmid instead of the 1:1 mass ratio, which is 5:1 molar ratio shown in the table in the methods. These large inaccuracies result in a misleading implied mechanism.

10. In Fig. 2e, do the negative values for PBMCs, HL-60, and CD8 T cells imply that adding the 3kb plasmid actually increased cell death for these cell types? Why is this not acknowledged in the manuscript?

11. The data shown in Fig. S3c and some in d are decidedly underwhelming. And surely not every cell type showed "***" statistical significance. For example, in d, there is no way the left and right data points are different for MCF7 cells. In that figure, there appear to be two green boxes on the right but only one green circle on the left. Be careful so that all the data actually show. I assume these are two repeats of the same experiment, but why would they range from 10 to 50% increase between repeated identical data points for HepG2 (in c)? The variability here seems far larger than the differences claimed.

Minor typos/clarity issue:

12. In the methods section (p. 8, line 143) is a spare word in the sentence and the last verb tense should match the previous two (should read: was aspirated, gently washed, AND DETACHED...) "Cells were maintained by splitting at ~70-90% confluency for which medium by was aspirated, cells gently washed with phosphate buffered saline (PBS, Sigma) and detaching them with 3 mL of a trypsin-EDTA solution (Sigma) for 3-5 min."

13. Is it "7-AAD" (as on p. 10, line 179 or "7AAD" (p. 1 of supplemental data, line 4 and in Fig. S3)?

Rebuttal Letter

First of all, we would like to thank the reviewers for all the insightful comments! We believe that we have addressed and implemented the changes suggested by the reviewers. We have also generated additional data for clarification, which greatly improved our work. In this document you will find the unedited comments from the reviewers (blue italic), our point-by-point responses (black), and revisions in the manuscript (red). Every figure has been updated and we have added additional supplementary figures. We have updated the references accordingly.

We are looking forward to hearing from you.

Sincerely,
Claudia Kutter

*From: jung-eun.lee@nature.com
Subject: Decision on manuscript COMMSBIO-19-1668-T
Date: 3 January 2020 at 22:06:49 CET
To: claudia.kutter@ki.se
Reply-To: jung-eun.lee@nature.com*

*** Please ensure you delete the link to your author home page in this e-mail if you wish to forward it to your coauthors ***

Dear Dr Kutter,

Your manuscript entitled "Successful delivery of large-size CRISPR/Cas9 vectors in hard-to-transfect human cells" has now been seen by 2 referees, whose comments are appended below. You will see from their comments copied below that while they find your work of considerable potential interest, they have raised quite substantial concerns that must be addressed. In light of these comments, we cannot accept the manuscript for publication, but would be interested in considering a revised version that addresses these serious concerns.

As you will see, while all reviewers find the method presented in this study potentially useful, they raised serious concerns regarding alternative possibilities. To address them, we ask you to please test whether the addition of small plasmids would improve the transfection efficiency of large DNA constructs at their optimal DNA concentration while transfection efficiency and cell viability are measured in a more accurate manner, as suggested by reviewers 1. We also ask that you test whether DNA mass also matters, as suggested by reviewer 2.

We hope you will find the referees' comments useful as you decide how to proceed. Should further experimental data or analysis allow you to address these criticisms, we would be happy to look at a substantially revised manuscript. However, please bear in mind that we will be reluctant to approach the referees again in the absence of major revisions. If the revision process takes significantly longer than six months, we will be happy to reconsider

your paper at a later date, as long as nothing similar has been accepted for publication at Communications Biology or published elsewhere in the meantime.

We are committed to providing a fair and constructive peer-review process. Do not hesitate to contact us if you wish to discuss the revision or if there are specific requests from the reviewers that you believe are technically impossible or unlikely to yield a meaningful outcome.

If you choose to resubmit your paper, we also ask that you ensure that your manuscript complies with our editorial policies. Specifically, please ensure that the following requirements are met, and any relevant checklists are completed and uploaded as a Related Manuscript file type with the revised article:

**Mandatory checklists:*

<https://www.nature.com/documents/nr-editorial-policy-checklist.pdf>

We provided the checklist on the 23rd November 2019 by email.

**If your study includes custom software, please also include the Software Checklist with your submission: <https://www.nature.com/authors/policies/Software.pdf>*

Software version used is included in Methods and Materials. No custom software was used.

**If your study includes characterization of chemical and biomolecular materials, the following checklist is needed: <https://www.nature.com/commsbio/submit/submission-guidelines#characterisation>*

N/A

Furthermore, your manuscript should comply with our format requirements, which are summarized on the following checklist:

<http://www.nature.com/authorguide/commsbio/commsbio-checklist.pdf>.

Our manuscript complies with the formatting requirements of Communication Biology

Data Availability

Complete information about our policies on the availability of data, materials and methods can be found on our policies page. All Communications Biology manuscripts must include a section titled "Data Availability" at the end of the Methods section or main text (if no Methods). See here for more information on this policy and a list of examples.

No data were generated that require deposition in public databases

We hope to receive your revised paper within six months; please let us know if you aren't able to submit it within this time so that we can discuss how best to proceed. If we don't hear from you, and the revision process takes significantly longer, we will close your file. In this event, we will still be happy to reconsider your paper at a later date, as long as nothing similar has been accepted for publication at Communications Biology or published elsewhere in the meantime.

Please do not hesitate to contact me if you have any questions or would like to discuss the required revisions further. Thank you for the opportunity to review your work.

Best regards,

*Jung-Eun Lee, PhD
Associate Editor, Communications Biology
One New York Plaza, Suite 4600
New York, NY 10004-1562
orcid.org/0000-0003-0184-3440
jung-eun.lee@nature.com*

Referee expertise:

Referee #1: Non-viral genome targeting

Referee #2: Effects of DNA length on electroporation efficiency

Reviewers' comments:

Reviewer #1 (Remarks to the Author):

This manuscript highlights the difficulties associated with delivering large DNA constructs by electroporation or chemical-based transfection. The authors demonstrate increased toxicity and reduced transfection efficiency as DNA length increases, a well-described phenomenon that poses significant challenges for several genome editing applications including the introduction of large multicistronic plasmids, as in the current study, but also with relevance for other plasmids of interest [1, 2]. The current manuscript suggests this effect may be partially overcome by co-transfection with plasmids of smaller size, with optimal benefit seen at a size of ~3 kb. While this would be an interesting and potentially useful finding, there are several technical limitations to the flow cytometric analysis and points of confusion with the described data that limit interpretation. A re-evaluation of the data to address these issues along with supplemental studies to identify the optimal DNA concentration used for transfection would help support the author's conclusions. In addition, the significance of the findings would be strengthened by demonstration of genome editing outcomes, especially given the author's focus on CRISPR-based genome editing. Finally, a discussion of alternative delivery methods such as RNP electroporation combined with a more complete discussion of the proposed mechanism would help frame this work.

Major points:

1. There are several issues with the current flow cytometric analysis. First, the bulk of the provided data is generated by gating the percentage of 7AAD+ events (“dead cells”) versus the percentage of GFP+7AAD- events (“GFP+ cells”) on a single plot, as exemplified in figure S1c. This strategy links these two populations so that when one goes up, the other must go down. In conditions with low viability, as in the left-most panel of figure 1d, the transfection efficiency will therefore appear artificially low because almost all events fall in the dead cell gate. This linkage may at least partially explain the strong positive correlation shown in figure 2f.

This issue can be overcome by measuring transfection efficiency as a percentage of live cells rather than total events. Second, the gating strategy includes a large population of cellular debris. This is visible as a second population with low forward-scatter in the example gating shown in figure S1a-b, and as multiple populations visible in the 7AAD+ gate of all example plots (e.g. figure 1d, S1c). Inclusion of this debris further confounds the current analysis, as each of these events contributes to the percentage of “dead cells” and, conversely, takes away from the percentage of “GFP+ cells”. This would be expected to overestimate the percentage of dead cells and further underestimate the transfection efficiency. Finally, the question of cellular viability is not adequately addressed by quantifying the relative percentage of live/dead cells at a single time point - this number will vary with mechanism of toxicity, time-course, proliferation, media change, and other factors. A more appropriate measurement is the number of live cells in each condition versus the number of live cells in an untreated control (which can be expressed as a percentage), or in the absence of an untreated control, directly comparing the absolute number of live cells between conditions. The 1st reference in this paper provides several excellent examples for addressing all of these concerns[1]. Specifically, please consider the following changes:

- a. Remove the population of low forward-scatter debris from all subsequent analyses. (Left-most population in figure S1a). Although debris could be related to toxicity, including these points confuses measurements of efficiency.*
- b. Evaluate transfection efficiency as percentage of GFP+ live cells (see Ref [1]- figure 1b and section “Flow cytometric analysis an measurement of viability”). An absolute count of GFP+ live cells should also be included to evaluate the absolute number of transfected cells.*
- c. Evaluate viability as the percentage of live cells versus an untreated control, or in the absence of an untreated control, directly comparing the absolute number of live cells between conditions (see Ref [1]- figure 1a and section “Flow cytometric analysis an measurement of viability”).*

The reviewer is correct that the leftmost cell population of the forward-side scatter flow cytometry plot will contain a mixture of debris and dead cells (see Fig. R1 below). Our results point towards a mechanism in which small vectors protect cells from cell death (potentially via the hypothesis put forward in Fig. 2g) during cell electroporation. Therefore, it would seem unreasonable to exclude the dead/debris population from our analysis. When an adherent cell dies in culture, the cell will detach itself from the plastic surface and would reside in the supernatant. The methods described in reference [1] (removing floating cells) would not account for the total number of dead cells. The reviewer is correct that when we do our gating on the same plot, we can expect an anticorrelation e.g. an increase of the GFP+7AAD- population and a decrease in the 7AAD+ population. To avoid misinterpretation of the data, we have included the following additional measures for each graph:

1. Percentage of GFP+ cells from all viable cells (placed in Supplementary Figures 1, 3-6, and 8)
2. Total amount of viable GFP+ cells acquired at a stable flow rate for 2 minutes. Subtle fluctuations in the flow rate (depending on temperature and pressure) might be accounted for by using flow-count beads. As we did not include flow-count beads in our experiments the presented numbers do not account for such fluctuations but are nevertheless comparable with each other. We commented on our strategy in the figure legend of Supplementary Fig. 1.
3. Small vector and no vector control data for viability measures (when applicable).

Fig. R1: Question about leftmost population in FS-SS flow cytometry plot. A549 cancer cells were electroporated with a titration of a CRISPR-GFP vector (15kb) from 0-7.5 $\mu\text{g}/1$ million cells. Above is the results without addition of a 3kb vector, and below after addition of a 3kb vector to the electroporation mixture. Irrespective of addition of the small vector, the leftmost population increases with increasing concentration of the large vector. This population will thus contain not only debris, but also recently dead cells. We have hypothesised that the effect of the small vector is to decrease the viable cells (mostly in the rightmost population) from dying/becoming debris, which is visible from this graph. Prior to this display, noise and doublets were gated away. The graph is representative of 4 independent experiments.

2. Some of the labeling and experimental details are unclear.

a. For figures 1e and S2f, show data as absolute % with and without plasmid, rather than absolute % increase, which is confusing to interpret as written. Please also incorporate changes described in point 1.

We have updated the respective figures to display absolute percentage instead of percentage increase and corrected the manuscript and figure legends accordingly. We have also added more supplemental information, including the suggestions made in point 1.

b. Please indicate the amount of each plasmid being added for each experiment (mass or moles). These details are essential for interpreting the data.

This data was already embedded in the materials and methods section. To retrieve this information easier, we added the requested information to Table 1 and also refer to it at several places in the main text and in the figure legends.

Table 1: Electroporation conditions used in each experiment (unless otherwise stated).

Cells	# of cells	µg GFP vector	µg small vector	Voltage	ms	pulses	Further optimized from manufacturer's settings
Huh7	10 ⁶	7.5	7.5	1000	40	2	Yes
HepG2	10 ⁶	2.5	7.5	1200	30	2	Yes
A549	10 ⁶	5	5	1230	30	2	No
HEK293	10 ⁶	5	5	1100	20	2	No
MCF7	10 ⁶	5	5	1100	30	2	No
HL60	10 ⁶	5	5	1350	35	1	No
PC3	10 ⁶	5	5	1450	10	3	No
SH-SY5Y	10 ⁶	5	5	1200	20	3	No
PBMCs	10 ⁶	5	5	2150	20	1	No
CD8 T cells	10 ⁶	5	5	2100	20	1	No

c. There are several points of confusion with the liposomal transfection experiments. The main text suggests similar benefits were seen with electroporation and liposomal-based transfection, supported by figure S3. As labeled, this figure appears to show that inclusion of the 3kb plasmid leads to increased transfection efficiency (figure S3c) but decreased viability (figure S3d). This contrasts with the increased viability seen with electroporation-based experiments. In addition, the explanatory text for figure S3 indicates that the 15kb “with” 3kb plasmid data should be on the left rather than right for figure S3c-d. Finally, it is unclear which of the comparisons in figure S3c-d the statistical significance is meant to apply to. Please consider the following changes:

- i. Correct the labeling and description for figure S3 to harmonize and accurately reflect data.*
- ii. If the current labeling of figure S3c-d is accurate, highlight the different results seen with electroporation and lipofection in the main text and provide some interpretation of these differences in your discussion.*
- iii. Indicate which comparisons the statistical significance applies to in figure S1c-d.*

The reviewer is correct that the results for lipofectamine are slightly different than for electroporation. Similar to electroporation, lipofectamine generally increased the number of GFP+ cells, but also increased the number of dead cells. We have added the following text to the revised version of the manuscript (starting line 136):

*“Similar to the results obtained after electroporation, the addition of the small vector enhanced transfection efficiencies in all of the tested cell types (**Supplementary Fig. 8**). However, in contrast to electroporation-based transfection, inclusion of the small vector in the liposomes slightly decreased the viability (**Supplementary Fig. 8**).”*

Our statistical comparisons in the original submission were performed by considering all replicates irrespective of cell line. As this representation has not been clear to both reviewers, we have created separate plots for each cell line.

Since we have added more data, we have also corrected the figure legends in the main figures and supplements.

3. The toxicity and transfection efficiency of a given DNA construct are highly sensitive to variations in DNA concentration [1, 2]. An alternative explanation for the effects seen in this study is that the molar excess of small plasmid outcompetes the larger plasmid, lowering the effective concentration and subsequent toxicity. If this example were true, one could achieve a similar effect by simply lowering the concentration of the large DNA construct. To address this concern, please first perform a titration to determine the optimal DNA concentration for viability and transfection efficiency in each cell line. An example of such an analysis is shown in figure 3 of Ref [1]. Next determine whether inclusion of the small plasmid at this optimal concentration shows the same benefit.

We have performed the suggested titration experiment by co-transfecting various amounts of the 15kb CRISPR-GFP vector in the presence (one defined amount) or absence of the small 3 kb vector. The data is summarized in Fig. 1e and Supplementary Fig. 3 and demonstrates that lowering the amount of the large construct also lowers both the percentage of 7AAD+ cells and GFP+ cells. Adding the small vector to the transfections with lower amounts of the large vector slightly decreased viability (but not significantly). In contrast, at higher amounts of the large vector, adding the small vector significantly increased the percentage of GFP+ cells and the number of viable cells in line with our model from Fig. 2g. The optimal amounts (5 μg of the 15kb CRISPR-GFP and 5 μg of 3kb small vector) was incidentally already used and reported in the previous version.

Supplementary Fig. 3: The small vector improves transfection efficiencies irrespective of the concentration of the large CRISPR-GFP vector. a-d, Line graphs demonstrate the percentage and number of viable, GFP+, and 7AAD+ (dead and debris) cells after co-transfection of varying amounts (0-7.5 μg) of a large CRISPR-GFP vector (15kb) without (grey, 0 μg) and with (green, 5 μg) a small vector (3kb) in A549 and MCF7 cells (n=4, mean +/- SEM). Two controls consisting of no vector and only the small vector represent the cell percentage or count when transfecting 0 μg of the large CRISPR-GFP vector (x-axes). Statistics: paired two-tailed t-test, *p < 0.05, **p < 0.01, ***p < 0.001.

4. The claims regarding consistent improvement in primary cells lines are not adequately demonstrated. From figure 2b, the maximum transfection efficiency appears to be ~2.5% for PBMCs and ~1% for CD8+ T cells with no difference in viability. While transfection efficiency is not directly comparable to genome editing outcomes (which were not examined, see point #7), knockout rates of >90% are routinely achieved with electroporation of pre-assembled Cas9:gRNA ribonucleoproteins (RNP) in these cell types. Within this context, the numbers

shown here would not be considered significant. Please rephrase claims regarding primary cells and provide discussion of alternatives such as RNP electroporation [2].

We have now corrected our statements in the manuscript and better describe the results obtained for the primary cell types. Starting at line 120: *“In contrast to adherent cells, cells in suspension had a very low transfection efficiency, which could probably be improved upon further optimization of the electroporation settings. However, co-transfection with the small vector did nevertheless improve the number of GFP+ cells (Fig. 2b,d, Supplementary Fig. 6) but did not increase cell viability (Fig. 2c,e).”*

We have also added the following text about RNP complexes to the revised manuscript (starting line 177): *“An alternative approach to our strategy has recently been published². Unlike transcribing the CRISPR/Cas9 system inside the host cell as we propose, a recombinant CRISPR/Cas9 ribonucleoprotein can be also produced *in vitro*. Co-transfection of large quantities of >1kb DNA with recombinant CRISPR/Cas9 ribonucleoprotein complexes markedly increased the viability of transfected cells². The caveat with this approach is that a recombinant protein has to be produced, and the ribonucleoprotein complexes must be electroporated immediately after formation. Compared to this strategy, our approach of adding a small vector to the transfection mixture is simpler, cheaper and less time-consuming.”*

Minor points:

5. The discussion of potential mechanisms is interesting but incomplete. What are the primary causes of DNA-mediated cellular toxicity and how would these be affected by DNA length? Is there evidence that entanglement in multiple pores is a major factor reducing transfection efficiency or viability? How would a 3kb plasmid reduce multi-pore entanglement of a 15kb plasmid? Could this mechanism also explain results seen with lipofection-based approaches? Please consider a more comprehensive discussion to frame these results and suggest future studies.

Besides from the mention of RNP complexes as an alternative, we have extended our final discussion as follows (starting line 161):

*“We postulate that small vectors protect the cell and nuclear membrane and act as a molecular lubricant. By co-travelling with small vectors, larger vectors may enter the cell without getting entangled in two or more open **membrane** pores, which ensures proper uptake and pore reclosure thereby preventing cell death⁷ (Fig. 2g). This model would explain the results we have obtained using electroporation of adherent cells, but it explains neither the poor viability of suspension cells’ nor why the small vector improves liposome transfection efficiency. Compared to the adherent cells, cells in suspension are generally smaller and have not undergone trypsinization. Trypsinization cleaves the bonds that stretch the membrane of the adherent cells and may thus result in a less compact and a more relaxed membrane structure. This may facilitate the formation of larger pores after electroporation, allowing better accessibility of small plasmids and thereby smoother membrane passage of larger vectors (Fig. 2g). The difference in cell death after co-transfection with small vectors may be further explained by different responses of DNA sensors triggering programmed cell death⁸. DNA sensors might be inert in response to small plasmids but highly active when sensing large plasmids. However, this phenomenon is*

speculative, and more work needs to be done in order to resolve the exact underlying mechanism.”

6. Most mammalian expression plasmids that incorporate Cas9, a selectable marker, and gRNA are in the 9kb range (<https://www.addgene.org/crispr/mammalian/>). The larger plasmids used here contain additional elements for viral packaging and integration that would only be relevant for use in packaging cell lines. While this difference may not change the results, it could be confusing or misleading for readers. Please adjust text to accurately reflect the available choices for plasmid-based delivery. For example, instead of 12-19kb, you could say 9-19kb or up to 19kb.

Thank you for the suggestion. We have changed the text accordingly throughout the manuscript.

7. There is an emphasis on CRISPR/Cas genome engineering throughout the paper, and yet there is no analysis of genome editing outcomes. If CRISPR/Cas9 is the example use-case, it would be beneficial to show how this method affects some readout of Cas9-mediated editing (e.g. indel formation, knockout efficiency, homology-directed-repair, etc.). In addition, there is no discussion of alternative methods, such as RNP electroporation, that are widely used and provide substantial increases in editing efficiency for many of the cell types included in this study [2].

In alignment with reviewer 1, point 4 above, we have added additional text concerning the usage of RNP complexes. Indeed, most of the GFP vectors used in this study are meant for genome engineering purposes e.g. the 15kb CRISPR-GFP vector (pLV hU6-sgRNA hUbC-dCas9-KRAB-T2a-GFP, Addgene# 71237). The expression of GFP is an easier, faster and scalable quantitative readout for defining optimal transfection efficiencies than e.g. quantifying gene expression through orthogonal molecular methods. Given the great number of experiments conducted in this study, we believe that measuring GFP fluorescence is sufficient measurement that supports our conclusions. The cells expressing GFP will also express the CRISPR components. Irrespective of transfection efficiencies, we believe that the efficiency for a specific modifications of gene expression by using the CRISPR/Cas system will be dependent on other factors, such as the design of guide RNAs.

References:

- 1. Lesueur, L.L., L.M. Mir, and F.M. Andre, Overcoming the Specific Toxicity of Large Plasmids Electrotransfer in Primary Cells In Vitro. Mol Ther Nucleic Acids, 2016. 5: p. e291.*
- 2. Roth, T.L., et al., Reprogramming human T cell function and specificity with non-viral genome targeting. Nature, 2018. 559(7714): p. 405-409.*

Reviewer #2 (Remarks to the Author):

Communications Biology Review

12/30/19

This manuscript contains a very interesting observation that the addition of “small” plasmids seems to increase transfection (done by any of several tested methods, although the significance for methods other than electroporation are unclear) of the very large DNA vectors needed to deliver CRISPR/Cas-encoding plasmids into several different experimentally relevant cell types. This very short and somewhat superficial manuscript is written well, and the figures are mostly clear.

The data shown appear to be done fairly well. The range of plasmid lengths is fairly extensive. The different cell types tested are also extensive. The observation is important and useful to the field and beyond. The phenomenon shines a light on just how little we understand the process of transfection. It also highlights that transfection has been optimized for in vitro work over a short period of time (here 24 hours). How cell death changes with time may be important but is ignored.

There are several issues related to quantitative evaluation that are lacking in the current manuscript:

1. Time following electroporation/transfection strongly affects cell death and maybe also transfection efficiency and varies per cell type. Both GFP fluorescence and cell viability were analyzed at 24 hours post transfection, but this one time point may be too short (or too long) to capture cell death for some cell types.

Per the reviewer’s request, we have performed a time course experiment 6h after transfection (day 0.25), day 1, 2, and 4. The improvements in transfection efficiencies upon adding the small vector were already seen after 6h and maintained throughout the time course until day 4. We have added the data to Fig. 1g and as a new Supplementary Fig. 5 and described our results in the manuscript (starting line 78):

“Additionally, GFP expression in cells that were transfected with a large CRISPR-GFP vector (15kb) in the presence of small (3kb) plasmids showed a stable enhancement in transfection efficiencies lasting for several days (6h to 4d after transfection) (Fig. 1g, Supplementary Fig. 5).”

Supplementary Fig. 5: The small 3kb vector improves cell viability and transfection efficiency for several days. a-d, Line graphs demonstrate the percentage and number of viable, GFP+, and 7AAD+ (dead and debris) cells after co-transfection of a large CRISPR-GFP vector (15kb) without (grey) and with (green) a small vector (3kb) in A549 and MCF7 cells at time points from 6h (0.25d) until day four after transfection (n=4, mean +/- SEM). Cell transfection controls with neither large nor small vector (black) and only the small 3kb vector alone (purple) are shown in a-b. Statistics: paired two-tailed t-test comparing the

large CRISPR-GFP vector with and without the small vector (green versus grey curves), *p < 0.05, **p < 0.01, ***p < 0.001.

2. What is the half-life of the GFP (which GFP variant is this?) encoded by the “15kb CRISPR-GFP vector?”

The 15kb CRISPR-GFP vector is pLV hU6-sgRNA hUbc-dCas9-KRAB-T2a-GFP (Addgene# 71237). The EGFP has an approximate half-life of about 26h (<https://horizondiscovery.com/FAQ/Horizon-Discovery/RNAi-Custom-RNA-Synthesis/What-is-the-half-life-of-eGFP>). Our time course analysis shows stable expression of GFP in cells over 4 days. We would not consider that the half-life of GFP has a strong influence on our observations and interpretations.

3. It seems that an important control experiment is missing of adding the equivalent increase of DNA mass (regardless of length) of just the 15kb plasmid to see whether increased DNA increases transfection and decreases cell death or whether the increased mass must only come from smaller plasmids. One would think that the effect of DNA length test done (shown in Fig. 1F) would test this possibility, but the same mass of 15 kb vector was added over each experiment and the same mass of “smaller” plasmid.

We have conducted an additional experiment with varying amounts of the large 15kb CRISPR-GFP vector with or without the small plasmid (see also reviewer 1 point 3). Increasing the amount of the larger vector increased the number of 7AAD+ cells (dead cells and debris) to a plateau of around 80%. The number of total GFP+ cells did not increase with increased large vector mass, and thus the effect we are observing from adding the small vector cannot be explained by an increased DNA mass. We have added the titration data to Fig. 1e and Supplementary Fig. 3.

4. Would NUMBER (rather than mass) of plasmids give the same result? I would like to see a column added to the table shown in the methods where moles of each of the “smaller” plasmids are also shown. The authors do not consider this possibility.

We have added an additional table (Table 2), with an extra column showing the molarity of each of the vectors. As we used equal mass of each of these vectors, the molarity will decrease with increasing vector size. As demonstrated from the titration data (see reviewer 1 and 2, point 3) it is the size of the vector rather than the number of molecules that affects the transfection efficiency. Increasing the moles of the 15 kb vector from 108 fmol (1 µg) to 810 fmol (7.5 µg) doubled the number of 7AAD+ cells (dead and debris), while the number of positively transfected cells remained the same. We have added the following text in the manuscript (starting line 63):

“Transfection of cells with increasing amounts of CRISPR-GFP vector (15kb, 0-7.5µg) did not result in an increase of GFP+ cells (Fig. 1e, Supplementary Fig. 3) but rather lead to a higher number of dead cells (Supplementary Fig. 3). Co-transfection of a fixed amount of the small vector (3kb, 5µg) increased the number of GFP+ cells consistently (4.3-fold change on average) and increased the number of viable cells (1.9-fold change on average). This suggests that the size but not the amount of the large vector affect transfection efficiencies.”

Table 2: Number of molecules of the different small vectors” used in Figure 1d

Small vector	size [bp]	mass [μg]	molarity [pmol]
pUC19-no-LacZ	1757	5	4.605
pUC19	2686	5	3.012
pBlueScript	2961	5	2.733
pH6HTC	3473	5	2.330
pH6HTC-STMN	4448	5	1.819
pH6HTC-PKM	5057	5	1.600
pH6HTC-CCT	5633	5	1.436
pH6HTC-CTCF	6209	5	1.303
pH6HTC-D9	6531	5	1.239

5. In Fig. 1a, the schematic showing transfection in tubes is overly simple. A schematic showing transfection, incubation, trypsinization, and analysis and including the temporal aspect of each of these steps is more useful than an overly detailed first step only.

We have updated Fig. 1a and the figure legend accordingly:

“Fig. 1: Transfection efficiency can be improved by co-transfecting large CRISPR vectors with small vectors. a, Schematic overview of the cell transfection setting. Electroporation-mediated transfection (lightning bolt) of a large CRISPR-GFP vector (15kb) without (above) and with (below) a small vector (3kb). Duration in days (d) and hours (h) for each experimental procedure is indicated.”

6. In Fig. 1c and S1h, please add size bars and define them in the legend.

We have added size bars to the microscopy images and defined them in the figure legends:

“Scale bar: 100 μm”

7. In Fig. 1e, the lines drawn are clearly not “best fit.” How were these data fit? If drawn merely as a visual guide, please state so. And do not fill in between “0” and the shortest length plasmid tested. One should never infer data not done.

We have updated our figures as per request of reviewer 1 and 2.

The line inserted into the previous Fig. 1e was merely a visual guidance to connect our separate measures. The graph showed the increase in percent when adding the small vector. Hence there is 0% increase when compared to adding no small vector. We do realize that this has been confusing and amended accordingly.

8. In Fig. 2, there are several issues:

(a) the “legend” for g that appears above the drawing would be improved if rather than a vector in purple the exact same length as the “small plasmid” being depicted for “CRISPR vector,” instead just put a purple line. These two circles are not the same size, as indicated wrongly in the figure.

(b) the sideways lines on the right side are not defined (and clutter the image). At first, I thought perhaps they were meant to convey increased negative charge (-) but that they had all turned sideways erroneously, but that is not a model mentioned, so it must mean something else.

(c) the “cell membrane” shown in G is confusing. In the text, the authors mention a model of the DNA entering nuclear pores. Is this actually the nuclear envelope instead of “cell membrane” as labeled? Either way, the model is not well thought through. If it is a cell membrane and what is shown is a small part of a larger cell, I do not understand why the membrane “ends” are rounded where no DNA is shown entering the cell. Is what is shown the electroporation case and there are holes (with no DNA)? Are we to assume that the bilayer membrane is depicted as the gray line where it is attached and therefore what is depicted is the whole cell or the whole nucleus? If it is supposed to be the nucleus, then the actual nuclear pores, which are not just discontinuations of nuclear envelope, should be shown but then “cell membrane cannot reseal” as a mechanism for cell death doesn’t make sense.

We have taken the reviewers concerns into consideration and updated our model in Fig. 2g. The lines referred to in question (b) was meant to indicate directionality and velocity. Since this was confusing, we have removed them in the updated version. Furthermore, we have added the following explanatory text regarding which membrane we are referring to:

“We postulate that small vectors protect the cell and nuclear membrane and act as a molecular lubricant. By co-travelling with small vectors, larger vectors may enter the cell without getting entangled in two or more open membrane pores, which ensures proper uptake and pore reclosure thereby preventing cell death⁷ (Fig. 2g).”

9. The net difference in length between the large 15kb and the small 3 kb plasmids is only 5-fold (15000 compared to 3000), but in Fig. 1B and Fig. 2, the small circle is drawn ~300-fold smaller. Furthermore, the number of small circles shown is at ~20:1 ratio to the large plasmid instead of the 1:1 mass ratio, which is 5:1 molar ratio shown in the table in the methods. These large inaccuracies result in a misleading implied mechanism.

Thank you for noting these inaccuracies in our figures. We have corrected them.

10. In Fig. 2e, do the negative values for PBMCs, HL-60, and CD8 T cells imply that adding the 3kb plasmid actually increased cell death for these cell types? Why is this not acknowledged in the manuscript?

There is indeed a slight decrease in viability in these three cell types. We have added the following explanatory text to the revised manuscript:

Starting at line 120: *“In contrast to adherent cells, cells in suspension had a very low transfection efficiency, which could probably be improved upon further optimization of the electroporation settings. However, co-transfection with the small vector did nevertheless improve the number of GFP+ cells (Fig. 2b,d, Supplementary Fig. 6) but did not increase cell viability (Fig. 2c,e).”*

Starting at line 164: *“This model would explain the results we have obtained using electroporation of adherent cells, but it explains neither the poor viability of suspension cells’ nor why the small vector improves liposome transfection efficiency. Compared to the adherent cells, cells in suspension are generally smaller and have not undergone trypsinization. Trypsinization cleaves the bonds that stretch the membrane of the adherent cells and may thus result in a less compact and a more relaxed membrane structure. This may facilitate the formation of larger pores after electroporation, allowing better accessibility of small plasmids and thereby smoother membrane passage of larger vectors (Fig. 2g).”*

11. The data shown in Fig. S3c and some in d are decidedly underwhelming. And surely not every cell type showed “***” statistical significance. For example, in d, there is no way the left and right data points are different for MCF7 cells. In that figure, there appear to be two green boxes on the right but only one green circle on the left. Be careful so that all the data actually show. I assume these are two repeats of the same experiment, but why would they range from 10 to 50% increase between repeated identical data points for HepG2 (in c)? The variability here seems far larger than the differences claimed.

The significance displayed in the original submission came from using all cell line replicates combined when comparing the efficiency and viability with and without co-transfection of the 3kb small vector. As this has been confusing, we have separated the data into the individual cell lines and generated new graphs (Supplementary Fig. 8). There is in general more variation in liposomal transfection compared to electroporation as variables such as liposome complex formation and cell density has a major effect on the outcome of the experiment.

Minor typos/clarity issue:

12. In the methods section (p. 8, line 143) is a spare word in the sentence and the last verb tense should match the previous two (should read: was aspirated, gently washed, AND DETACHED...) “Cells were maintained by splitting at ~70-90% confluency for which medium by was aspirated, cells gently washed with phosphate buffered saline (PBS, Sigma) and detaching them with 3 mL of a trypsin-EDTA solution (Sigma) for 3-5 min.”

The reviewer is correct that this sentence contained mistakes and was hard to read. We have updated it to the following (line 201):

“Cells were split at ~70-90% confluency by aspirating the medium, gently washing with phosphate buffered saline (PBS, Sigma) and detaching them with 3 mL of a trypsin-EDTA solution (Sigma) for 3-5 min.”

13. Is it “7-AAD” (as on p. 10, line 179 or “7AAD” (p. 1 of supplemental data, line 4 and in Fig. S3)?

It is 7AAD, and we have changed it accordingly throughout the manuscript.

REVIEWERS' COMMENTS:

Reviewer #1 (Remarks to the Author):

The authors have provided additional analyses including most of the requested modifications. Importantly, they have now evaluated transfection efficiency in viable cells and show absolute numbers of GFP+ cells over a range of DNA concentrations. With inclusion of this data, the author's conclusions are generally well-supported. The finding that co-transfection with a small piece of DNA improves delivery of large plasmids (>6kb) is interesting and potentially useful for the scientific community. We are overall supportive of publication but highlight several remaining points.

Major points:

1) The discussion of RNP editing implies this is a recent approach described in 2018, which is inaccurate and suggests a gap in knowledge. This method was described for Cas9-based editing of human T cells in 2015 (Schumann et al., PNAS 2015) and is the de facto standard for human blood cell editing (PBMCs, T cells, HSCs, B cells, erythroid progenitors, etc). This is true in the research setting and for clinical applications. The percentage of cells successfully edited with RNP electroporation is routinely >90%, in contrast to the ~5-20% transfection shown here (again, no editing outcomes were shown in this manuscript). Cas9 protein and de novo synthesized gRNA are relatively inexpensive and widely available, both at research and GMP-grade. In comparison to cloning, it is certainly arguable which approach would be "simpler, cheaper, and less time-consuming", not considering that RNP-based editing would be more effective. DNA delivery remains important, however, for a variety of alternative cargoes. The authors may be better served describing this as a general tool for DNA delivery in hard-to-transfect cell types, rather than focusing the manuscript on CRISPR-Cas9.

2) Page 8: "Co-transfection of large quantities of >1kb DNA with recombinant CRISPR/Cas9 ribonucleoprotein complexes markedly increased the viability of transfected cells"

a. Better to leave this sentence out as it is misleading. The real benefit of RNP is that it works better than plasmid-based delivery in terms of genome editing efficiency and viability. The key point is that plasmid DNA encoding Cas9 is no longer required for Cas9/gRNA delivery when RNPs are used.

Minor points:

3) Abstract: "CRISPR/Cas genome engineering relies on the delivery of large size vectors (9-19kb) into human cells resulting in low transfection efficiency and cell viability."

a. Statement about CRISPR-Cas reliance on plasmid-based delivery is overly broad. RNP delivery and viral delivery are standard, and better, for a wide variety of human cell types and clinical applications.

4) Page 1: "Viral-mediated delivery gives the highest transfection efficiencies but entails tremendous biological safety issues and ethical concerns when used in research or in the clinic".

a. Unclear what this statement means. Viral delivery is referred to as transduction. If meant to comment on genome editing outcomes, this is cell-type and context dependent, often RNP or viral delivery is better. So far, the current generation of engineered viruses used in clinic appear to have an overall good safety profile. What are the specific ethical concerns?

5) Page 1: "We overcame limitations of current clinically approved electroporation methods by adding appropriate amounts of small (~ 3kb) to large (9-15kb) vectors, which resulted in a significant increase of transfection efficiency and cell viability (Fig. 1a)."

a. What is meant by clinically approved electroporation methods? Are cGMP-compatible electroporation device, SOP, or reagents being used in this study? The authors should be more specific about what they mean in this context.

6) The authors have chosen to maintain the original gating strategy in the primary figures, which provides a readout of transfection efficiency in total events (including subcellular debris, dead cells, and live cells). With this strategy, the transfection efficiency will appear to increase with improvements in viability due to higher percentage of live events, regardless of changes to actual transfection efficiency. This has been relegated to a minor point as the conclusion of increased transfection efficiency is now supported by gating on viable cells in Supplementary Figures 1-5. However, the magnitude of change demonstrated in the main figures remains unreliable and is probably over-estimated. As an example from page 2 of the main text, a 6.8-fold increase in transfection efficiency is estimated for the 3kb plasmid based on Figure 1d (~3.1% -> 21.4%), whereas gating on viable cells in Supplemental Figure 1f demonstrates <3-fold is a more accurate estimation (~20%->~60%). All the fold-changes shown for different cell types in Figure 2d are similarly over-estimated in comparison to those provided in Supplemental Figure 6c. Ideally, the authors would acknowledge that this gating strategy may cause this artifact.

7) The choice to include subcellular debris in the % of dead or non-viable cells will amplify effects on viability as each dead cell will produce numerous subcellular particles. Again, this should be acknowledged.

8) Page 2: "Co-transfection of a fixed amount of the small vector (3kb, 67 5µg) increased the number of GFP+ cells consistently (4.3-fold change on average) and increased the number of viable cells (1.9-fold change on average). This suggests that the size but not the amount of the large vector affect transfection efficiencies."

a. This statement is inaccurate. Supplemental Figure 3c clearly shows increasing transfection efficiency with increasing concentration of large vector, as has been demonstrated by numerous other studies. The absolute number of GFP+ cells does not go up because there is a compensatory decrease in the number of viable cells. Suggest removing the last sentence.

9) Page 5: "Overall, we noticed that the increased percentage of positively transfected cells correlated highly with the percentage of viable cells (Fig. 2f), suggesting that the mode of action of the small vector is to improve viability of positively transfected cells".

a. Again, this correlation is expected with the chosen gating strategy, which directly links viability and transfection efficiency. Furthermore, there is some evidence arguing against the conclusion regarding mode of action. There appears to be significantly increased transfection efficiency even at the lowest dose of large plasmid (Supp Fig 3c, 1µg/10⁶ cells) while there is no effect on viability (Supp Fig 3a, 1µg/10⁶ cells). Several of the alternative cell lines tested show a similar pattern (Fig 2d vs. 2e).

10) The discussion of mechanism still centers on the concept of small vectors acting as a "molecular lubricant" for larger vectors. Yet there is no explanation for how a small vector would "lubricate" a pore, including in the noted reference (Ref 7). The authors now clarify that this model could not explain many of the results, so why it is proposed as a potential mechanism?

Reviewer #2 (Remarks to the Author):

I am impressed with the new data, and more importantly the results of the new data, in the revised version of the manuscript. My previous major concerns regarding conclusions drawn from only a single time point (24 hours post-transfection) and the need for separating the contribution of DNA vector mass from the contribution of length of vector to the observed improved cell transfection and

decreased cell death have both been addressed well.

Although I still feel the manuscript falls somewhat on the superficial or preliminary side and some of the data (luckily none of the key data) are not as convincing as the authors insist, the novelty of the observations and the potential usefulness of the results for others in the field combine to overcome the weaknesses.

Rebuttal Letter

In this document you will find the unedited comments from the reviewers (blue italic), our point-by-point responses (black), and revisions in the manuscript (red).

Sincerely,
Claudia Kutter

*** Please ensure you delete the link to your author home page in this e-mail if you wish to forward it to your coauthors ***

Dear Dr Kutter,

Your manuscript entitled "Successful delivery of large-size CRISPR/Cas9 vectors in hard-to-transfect human cells" has now been seen again by our referees, whose comments appear below. In light of their advice I am delighted to say that we are happy, in principle, to publish a suitably revised version in Communications Biology under the open access CC BY license (Creative Commons Attribution v4.0 International License).

We ask you to address the minor comments of reviewer #2 and to update the Reporting summary (date, blank in Statistics section).

We therefore invite you to revise your paper one last time to address the remaining concerns of our reviewers. At the same time we ask that you edit your manuscript to comply with our format requirements and to maximise the accessibility and therefore the impact of your work.

** Please see the attached document for editorial requests for the final version (.docx file).*

** Please review our final submission file checklist to ensure all necessary files are present with your final submission and to avoid delays in accepting your manuscript.*

It is important that you pay careful attention to the requests in these documents to avoid a delay in formal acceptance of the article.

Open access

Communications Biology is a fully open access journal. Articles are made freely accessible on publication under a CC BY license (Creative Commons Attribution 4.0 International License). This license allows maximum dissemination and re-use of open access materials and is preferred by many research funding bodies. For further information about article processing charges, open access funding, and advice and support from Nature Research, please visit our website.

Please use the following link to upload your revised files:

We hope to hear from you within two weeks; please let us know if the process may take longer.

Congratulations on an excellent paper!

Best regards,

Jung-Eun Lee, PhD
Associate Editor, Communications Biology
One New York Plaza, Suite 4600
New York, NY 10004-1562
orcid.org/0000-0003-0184-3440
jung-eun.lee@nature.com

PS: At acceptance, the corresponding author will be required to complete an Open Access Licence to Publish on behalf of all authors, declare that all required third party permissions have been obtained and provide billing information in order to pay the article-processing charge (APC) via credit card or invoice. Please note that your paper cannot be sent for typesetting to our production team until we have received these pieces of information; **therefore, please ensure that you have this information ready when submitting the final version of your manuscript.**

REVIEWERS' COMMENTS:

Reviewer #1 (Remarks to the Author):

The authors have provided additional analyses including most of the requested modifications. Importantly, they have now evaluated transfection efficiency in viable cells and show absolute numbers of GFP+ cells over a range of DNA concentrations. With inclusion of this data, the author's conclusions are generally well-supported. The finding that co-transfection with a small piece of DNA improves delivery of large plasmids (>6kb) is interesting and potentially useful for the scientific community. We are overall supportive of publication but highlight several remaining points.

Thank you for your support!

Major points:

1) The discussion of RNP editing implies this is a recent approach described in 2018, which is inaccurate and suggests a gap in knowledge. This method was described for Cas9-based editing of human T cells in 2015 (Schumann et al., PNAS 2015) and is the de facto standard for human blood cell editing (PBMCs, T cells, HSCs, B cells, erythroid progenitors, etc). This is true in the research setting and for clinical applications. The percentage of cells successfully

edited with RNP electroporation is routinely >90%, in contrast to the ~5-20% transfection shown here (again, no editing outcomes were shown in this manuscript). Cas9 protein and de novo synthesized gRNA are relatively inexpensive and widely available, both at research and GMP-grade. In comparison to cloning, it is certainly arguable which approach would be “simpler, cheaper, and less time-consuming”, not considering that RNP-based editing would be more effective. DNA delivery remains important, however, for a variety of alternative cargoes. The authors may be better served describing this as a general tool for DNA delivery in hard-to-transfect cell types, rather than focusing the manuscript on CRISPR-Cas9.

We have rephrased the sentence to the following, and added the additional reference (line 146):

“An alternative approach to our strategy has **previously** been published^{2,9}.”

Introducing foreign nucleic acids via viral delivery (transduction) and RNP complexes are alternative approaches, which we have discussed in the manuscript. Many laboratories will rely on plasmid delivery and we provide an improved version for plasmid-based approaches.

We have also added that RNP complexes are currently a better choice for transfecting primary blood cells and now reads (line 146):

“Unlike transcribing the CRISPR/Cas9 system inside the host cell as we propose, a recombinant CRISPR/Cas9 ribonucleoprotein can be formed in vitro **prior to electroporation. This approach yielded higher transfection efficiencies in primary blood cells than we report.**”

*2) Page 8: “Co-transfection of large quantities of >1kb DNA with recombinant CRISPR/Cas9 ribonucleoprotein complexes markedly increased the viability of transfected cells”
a. Better to leave this sentence out as it is misleading. The real benefit of RNP is that it works better than plasmid-based delivery in terms of genome editing efficiency and viability. The key point is that plasmid DNA encoding Cas9 is no longer required for Cas9/gRNA delivery when RNPs are used.*

We have removed the sentence as suggested by the reviewer and now reads (line 146):

“Unlike transcribing the CRISPR/Cas9 system inside the host cell as we propose, a recombinant CRISPR/Cas9 ribonucleoprotein can be formed in vitro **prior to electroporation. This approach yielded higher transfection efficiencies in primary blood cells than we report.**”

Minor points:

3) Abstract: “CRISPR/Cas genome engineering relies on the delivery of large size vectors (9-19kb) into human cells resulting in low transfection efficiency and cell viability.”

a. Statement about CRISPR-Cas reliance on plasmid-based delivery is overly broad. RNP delivery and viral delivery are standard, and better, for a wide variety of human cell types and clinical applications.

We agree with the reviewer that the statement is too broad. After adjusting the abstract as suggested by the editor, this sentence as such no longer exists (line 29):

“**Since the vectors encoding for the components necessary for CRISPR/Cas genome engineering are always large (9-19kb), they result into low transfection efficiency and cell viability, and thus subsequent selection or purification of positive cells is required.**”

4) Page 1: “Viral-mediated delivery gives the highest transfection efficiencies but entails tremendous biological safety issues and ethical concerns when used in research or in the clinic”.

a. Unclear what this statement means. Viral delivery is referred to as transduction. If meant to comment on genome editing outcomes, this is cell-type and context dependent, often RNP or viral delivery is better. So far, the current generation of engineered viruses used in clinic appear to have an overall good safety profile. What are the specific ethical concerns?

We agree with the reviewer that the statement is too broad and changed it to (line 47):

“Viral-mediated delivery (transduction) leads to the highest transfection efficiencies but requires higher biosafety level laboratory settings and ethical approval when used in research or in the clinic⁵”

5) Page 1: “We overcame limitations of current clinically approved electroporation methods by adding appropriate amounts of small (~ 3kb) to large (9-15kb) vectors, which resulted in a significant increase of transfection efficiency and cell viability (Fig. 1a).”

a. What is meant by clinically approved electroporation methods? Are cGMP-compatible electroporation device, SOP, or reagents being used in this study? The authors should be more specific about what they mean in this context.

We agree with the reviewer that the statement is too broad and changed it to (line 50):

“We overcame limitations of current electroporation-based transfections by adding appropriate amounts of small (~ 3kb) to large (9-15kb) vectors...”

6) The authors have chosen to maintain the original gating strategy in the primary figures, which provides a readout of transfection efficiency in total events (including subcellular debris, dead cells, and live cells). With this strategy, the transfection efficiency will appear to increase with improvements in viability due to higher percentage of live events, regardless of changes to actual transfection efficiency. This has been relegated to a minor point as the conclusion of increased transfection efficiency is now supported by gating on viable cells in Supplementary Figures 1-5. However, the magnitude of change demonstrated in the main figures remains unreliable and is probably over-estimated. As an example from page 2 of the main text, a 6.8-fold increase in transfection efficiency is estimated for the 3kb plasmid based on Figure 1d (~3.1% -> 21.4%), whereas gating on viable cells in Supplemental Figure 1f demonstrates <3-fold is a more accurate estimation (~20%->~60%). All the fold-changes shown for different cell types in Figure 2d are similarly over-estimated in comparison to those provided in Supplemental Figure 6c. Ideally, the authors would acknowledge that this gating strategy may cause this artifact.

We decided to keep the original gating strategy in the manuscript because for our study it is important to assess the fate of the entire cell population after electroporation. If we had gated away dead cells, the percentage of transfection efficiency would be artificially high, as the cells that died from the electroporation will be disregarded. We have acknowledged that our gating strategy would not account for one cell turning into several pieces of debris (Supplementary Fig. 1 legend):

“The 7AAD gate will contain both recently dead cells as well as subcellular debris, which may distort the measurement of actual viability as one dead cell may break down into several debris.”

7) The choice to include subcellular debris in the % of dead or non-viable cells will amplify effects on viability as each dead cell will produce numerous subcellular particles. Again, this should be acknowledged.

We have added the following text to our gating strategy description in Supplementary Fig. 1:

“The 7AAD gate will contain both recently dead cells as well as subcellular debris, which may distort the measurement of actual viability as one dead cell may break down into several debris.”

8) Page 2: “Co-transfection of a fixed amount of the small vector (3kb, 67.5 μg) increased the number of GFP+ cells consistently (4.3-fold change on average) and increased the number of viable cells (1.9-fold change on average). This suggests that the size but not the amount of the large vector affect transfection efficiencies.”

a. This statement is inaccurate. Supplemental Figure 3c clearly shows increasing transfection efficiency with increasing concentration of large vector, as has been demonstrated by numerous other studies. The absolute number of GFP+ cells does not go up because there is a compensatory decrease in the number of viable cells. Suggest removing the last sentence.

The total count of GFP+ cells does not increase with increasing plasmid concentration (Supplementary Fig. 3d). However, as the reviewer mentions when excluding dead cells from the total cell population (Supplementary Fig. 3c), the percentage of GFP+ cells increases with increasing large-size plasmid concentrations. This would suggest that higher concentrations of large-size plasmid increase the number of dead cells, as we observed in Supplementary Fig. 3a-b. See also point 6.

9) Page 5: “Overall, we noticed that the increased percentage of positively transfected cells correlated highly with the percentage of viable cells (Fig. 2f), suggesting that the mode of action of the small vector is to improve viability of positively transfected cells”.

a. Again, this correlation is expected with the chosen gating strategy, which directly links viability and transfection efficiency. Furthermore, there is some evidence arguing against the conclusion regarding mode of action. There appears to be significantly increased transfection efficiency even at the lowest dose of large plasmid (Supp Fig 3c, 1 μg/10⁶ cells) while there is no effect on viability (Supp Fig 3a, 1 μg/10⁶ cells). Several of the alternative cell lines tested show a similar pattern (Fig 2d vs. 2e).

See point 10.

10) The discussion of mechanism still centers on the concept of small vectors acting as a “molecular lubricant” for larger vectors. Yet there is no explanation for how a small vector would “lubricate” a pore, including in the noted reference (Ref 7). The authors now clarify that this model could not explain many of the results, so why it is proposed as a potential mechanism?

We have removed the term “molecular lubricant”.

The model is based on Stewart *et al.*, Chem. Rev., 2018 (Ref 7) description of large plasmids getting stuck in the membranes of cells and our actual data. It is true that the model does not describe every potential cellular scenario, which we have also stated as such in the manuscript. At present, this is the best model explaining our findings. When future mechanistic studies are performed, we may need to revisit our model.

Reviewer #2 (Remarks to the Author):

I am impressed with the new data, and more importantly the results of the new data, in the revised version of the manuscript. My previous major concerns regarding conclusions drawn from only a single time point (24 hours post-transfection) and the need for separating the contribution of DNA vector mass from the contribution of length of vector to the observed improved cell transfection and decreased cell death have both been addressed well.

Although I still feel the manuscript falls somewhat on the superficial or preliminary side and some of the data (luckily none of the key data) are not as convincing as the authors insist, the novelty of the observations and the potential usefulness of the results for others in the field combine to overcome the weaknesses.

Thank you for your support!

*** See Nature Research's author and referees' website at www.nature.com/authors for information about policies, services and author benefits*

COVID 19 and impact on peer review

As a result of the significant disruption that is being caused by the COVID-19 pandemic we are very aware that many researchers will have difficulty in meeting the timelines associated with our peer review process during normal times. Please do let us know if you need additional time. Our systems will continue to remind you of the original timelines but we intend to be highly flexible at this time.

COMMSBIO - This email has been sent through the Springer Nature Tracking System NY-610A-NPG&MTS

Confidentiality Statement:

This e-mail is confidential and subject to copyright. Any unauthorised use or disclosure of its contents is prohibited. If you have received this email in error please notify our Manuscript Tracking System Helpdesk team at <http://platformsupport.nature.com> .

Details of the confidentiality and pre-publicity policy may be found here <http://www.nature.com/authors/policies/confidentiality.html>

Privacy Policy | Update Profile

DISCLAIMER: This e-mail is confidential and should not be used by anyone who is not the original intended recipient. If you have received this e-mail in error please inform the sender and delete it from your mailbox or any other storage mechanism. Springer Nature America, Inc. does not accept liability for any statements made which are clearly the sender's own and not expressly made on behalf of Springer Nature America, Inc. or one of their agents. Please note that neither Springer Nature America, Inc. or any of its agents accept any responsibility for viruses that may be contained in this e-mail or its attachments and it is your responsibility to scan the e-mail and attachments (if any).